# Revolutionizing Event Detection: A novel Prompt-Driven Method Enhanced by Retrieval-Augmented Paradigm

## Abstract

Event Detection (ED) task involves extracting event triggers from sentences and classifying them into predefined event types. While large language models (LLMs) have become widely adopted across various NLP tasks, their application to ED remains relatively unexplored. All existing LLM-based approaches follow a traditional prompt-based paradigm, which requires designing distinct prompts for each event type. This strategy, however, suffers from a fundamental limitation: as the number of event types grows, the number of prompts needed increases linearly, resulting in significant manual effort and computational costs. To overcome this limitation, we propose a novel approach that integrates a retrieval-augmented mechanism with a redesigned cascading prompt-based framework. Specifically, the prompt-based component is employed to extract candidate triggers, while the retrieval-augmented module applies heuristic filtering strategies to coarsely eliminate irrelevant candidates. In addition, we put forward an innovative automated prompt-design method to accurately match valid triggers with their corresponding event types based on retrieved information. Experimental results on ACE-05 benchmark demonstrate the state-of-the-art performance under our scheme. Furthermore, the approach remains highly effective when using lightweight LLMs, indicating its strong potential for efficient large-scale data processing. This capability may have profound implications and become a fundamental work for future research.

## 1 Instruction

Nowadays, Event Detection (ED) Ahn (2006) Wadden et al. (2019) is playing an increasingly vital role with the rapid expansion of social media and various other forms of content proliferating on the Internet. The Event Detection task involves identifying words that explicitly signal the occurrence of an event within a sentence, known as event triggers, and classifying them into predefined event types. Until now, current research methodologies in event detection remain predominantly focused on supervised learning. For instance, Liao et al. (2021) and Liu et al. (2023) applied contrastive learning to better capture the discriminative features of trigger words. Abdi (2024) improved recognition performance by incorporating external corpora. Kan et al. (2024) proposed a novel pre-training strategy that enhances the ability of BERT-based models to identify triggers. However, all of the supervised learning studies require large volumes of annotated data for effective training. This dependency entails significant manual effort for data labeling. Moreover, such models exhibit inherent limitations in comprehension capacity, resulting in performance bottlenecks - particularly in the challenging environments that require effective knowledge transfer and strong generalization.

By comparison, the existing LLM-based approaches, notwithstanding their scarcity, are all underpinned by a shared foundational methodology. Gao et al. (2023) and Ma et al. (2024) adopted strategies that rely on

careful prompt design. These prompts primarily consist of event-type definitions and contextual examples. The definitions describe what each event type means: for instance, "Business:Start-Org" is defined as "Involves an organizational business, recording the formation of an organization." This approach enables LLMs to better comprehend the semantic information of structured triggers and their relevant event type. A notable drawback, however, is the need to pre-write a large number of prompts to cover all event types in advance. Shiri et al. (2024) proposed to utilize cosine similarity combined with FAISS to automatically generate examples, which are then used to incorporate into prompts as contextual few-shot demonstrations. Nevertheless, this approach focuses exclusively on sentence-level contextual semantics, overlooking the specific semantics of the extracted triggers themselves, which could lead to suboptimal performance.

Based on the aforementioned analysis, our study still focuses on LLMs, the current mainstream research direction, and proposes a revolutionary paradigm to address the above existing issues. Our pipeline contains three main modules: Candidate Trigger Extractions, Chroma Database Constructions, as well as Trigger Judgments and Classification. The Candidate Trigger Extraction module employs a novel multi-step cascading prompt strategy. By leveraging an iterative refinement process based on prior extractions, this module enables the discovery of candidate triggers that prior steps failed to detect. It consists of three sub-modules: the Basic Extraction Module, the Supplementary Extraction Module, and the Iterative Refinement Module. These three components(Section 3.1) employ systematic and logical design principles, achieving architectural consistency while facilitating LLM-based comprehensive extraction. Subsequently, our work proposes a retrieval-based method by leveraging the training dataset to construct two chroma databases, which are founded on word-level contextual semantics and sentence-level semantics, respectively. To improve the retrieval quality, arguments and non-ground-truth candidate triggers are incorporated as negative samples, thereby increasing the precision of candidate filtering. This is further supported by a parallel strategy that integrates sentence-level and contextual word-level similarities from both chroma databases to form a comprehensive score. Together, these components form the foundation for identifying and excluding non-trigger candidates accurately and subsequently judged by a three-stage filtering pipeline: an initial threshold filter (set at 0.7), followed by two heuristic rules, and an iterative automated prompt design process powered by Qwen3-1.7B, which finally determines the results. The experimental results demonstrate the effectiveness of our approach on the ACE-05 benchmarkLu et al. (2021), outperforming previous state-of-the-art methods by $2.78\%$ in trigger extraction and $4.74\%$ in event type classification. Notably, the entire process requires only 9 prompts with a total of pre-designed 44 demonstrations, highlighting the efficiency and practicality of our method. In addition, we need to mention that the evaluation criterion we use is **strict matching F1-score**. To the best of our knowledge, only one latest state-of-the-art work Ma et al. (2024) adopts the same evaluation criterion. Besides surpassing their performance, our experiments reveal that when deployed on lightweight LLMs, even Qwen3-1.7B, our framework can also achieve competitive performance. This finding has important implications for efficiently processing large-scale and dynamically-updated data, which may inspire further research in low-resource and time-sensitive scenarios.

Overall, the main contributions can be summarized as follows:

- We propose a novel LLMs-based approach for event detection (ED), which consists of a multi-step extraction module for candidate extraction and a fine-grained retrieval-augmented mechanism. The experimental results demonstrate the effectiveness of our proposed method.

- We further introduce an automated prompt-design approach driven by large language models (LLMs). By leveraging LLMs to judge the final outcomes iteratively, our method achieves notable filtering improvements, demonstrating the efficacy of this innovative strategy.

- We show that our framework, when deployed on lightweight LLMs (such as Qwen3 series), achieves performance beyond the state-of-the-art results implemented in pioneer models (GPT-4.0). The design principles underlying our approach pave the way for efficiently processing real-time data under resource restrictions, which may have profound meaning for future research.

## 2    RELATED WORK

There is a growing body of research applying supervised learning to Event Detection (ED) task. DNR Liao et al. (2021) employs a contrastive learning scheme with a MixSpan strategy to improve the accuracy of boundary detection in event extraction. CorED Sheng et al. (2022) proposes to learn latent relationships among event types. They capture the underlying connections between event types through the training process. Additionally, a masked attention mechanism is employed to enhance the model's focus on masked triggers, further improving the overall performance. PromptLoc Liu et al. (2023) introduces a contrastive learning strategy regularized by Gaussian distribution-based distance. It also adopts a self-correction mechanism based on MC dropout to increase the model's confidence in correct predictions. TAE Guan et al. (2023) observes that existing methods primarily focus on learning target triggers' features for extraction, while overlooking the potential of semantic relationships among all tokens in the sentence. Their work conducts experiments to capture meaningful associative knowledge and achieves competitive results. Abdi (2024) employs an ontology corpus and aligns event relations with ontological relations via optimal transport. LFDe framework Kan et al. (2024) introduces a novel pre-training strategy that equips the bert-based models with the ability to identify event triggers effectively and efficiently.

Research using LLMs for event detection remains relatively limited. Gao et al. (2023) utilizes fine-grained enhanced instructions, involving designing tailored prompts for specific event types to enhance LLMs' overall comprehension on triggers. Shiri et al. (2024) proposes an automated technique for constructing in-context examples. Through using FAISS retrieval, their work identifies training examples with high sentence-level semantic similarity to inferred samples, which are then integrated into the prompt. However, a limitation of this approach is that similarity operates mainly at the sentence level but fails to capture trigger-specific contextual semantics. STAR Ma et al. (2024) proposes a structure-to-text data generation framework accompanied by a self-refinement strategy. The synthesized data are combined with the original dataset to form k representative examples per event type, incorporated into prompts sequentially as demonstrations to promote event identification and classification. However, the length of the prompts and their constituent demonstrations scales linearly with the number of event types increases. Nearly all existing methods fundamentally rely on conventional prompt engineering strategies like Ma et al. (2024). As the number of predefined event types grows, these methods require considerable manual efforts, posing a significant constraint to their scalability and generalization to a broader set of event types.

## 3    METHODOLOGY

The pipeline of our proposed method, REVO-ED, is presented in Figure 1. It comprises three core modules: *Prompts of Candidate Trigger Extractions*, *Chroma Database Construction*, *Triggers Judgments and Classifications*. *Prompts of Candidate Trigger Extractions* is designed to extract potential candidate triggers through crafted prompts. It consists of three sub-modules: (1) the *Basic Extraction Module*, which identifies general, special, and pronoun-based triggers in straightforward semantic contexts; (2)the *Supplementary Extraction Module*, which will recover potential triggers missed by the *Basic Extraction Module*; (3) the *Iterative Refinement Module*, which handles complex semantic and syntactic structures that still remain unaddressed after the previous two stages. *Chroma Database Construction* aims to construct two vector databases using the training set. The final pipeline *Triggers Judgments and Classifications* employs filtering procedures to determine the final triggers and their event types.

### 3.1    PROMPT EXTRACTION OF CANDIDATE TRIGGERS

There has been some researches Li et al. (2023) Chen et al. (2024) demonstrates that large language models have limited capabilities in information extraction tasks. This phenomenon is particularly evident in event detection, where the complex semantic environments surrounding trigger words pose significant chal-

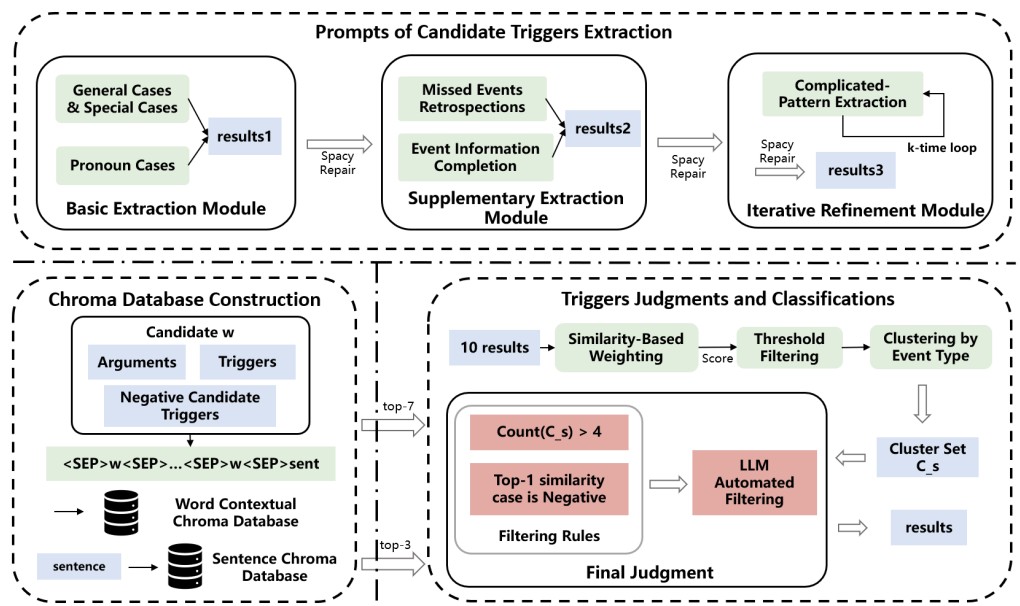

Figure 1: Pipeline of REVO-ED.

lenges for accurate identification through LLMs. From a syntactic perspective, triggers can include verbs (e.g., "kill", "sentence"), nouns (e.g., "meeting", "summit"), pronouns (e.g., "it", "this"), and multi-word expressions (e.g., "World War Two", "smash through"). To effectively handle this diversity, we first extract candidate triggers aimed at achieving high recall, covering nearly all ground-truth labels, and filter out negative samples subsequently. Note that the extracted results will inevitably introduce arguments and other irrelevant words. In addition, to enhance the LLMs' capacity for comprehending semantic knowledge of candidates and ensure their effective extraction, we propose an innovative hierarchically cascading prompt approach. In this framework, the extraction process of each succeeding sub-module is critically dependent on the accumulated results from preceding sub-modules, as described in 3.1.1, 3.1.2, 3.1.3.

### 3.1.1 BASIC EXTRACTION MODULE

The Basic Extraction Module is designed to handle three category cases: "general cases," "special cases," and "pronoun cases." "General cases" refer to extract candidates in simple semantic scenarios, such as the sentence: "Giuliani, 58, proposed to Nathan, a former nurse, during a November business trip to Paris five months after he finalized his divorce from Donna Hanover after 20 years of marriage." In this case, the extracted triggers should be: "proposed, trip, finalized, divorce, marriage". "Special cases" involve more complex linguistic constructs, including candidates enclosed in quotation marks, verb phrases, noun phrases, and instances where no trigger is existed. For example, from the sentence: "As the US-led coalition troops are reportedly **thrusting into** Baghdad and the second Iraqi city of Basra, Blair and Bush agreed there would be a 'vital **role**' for the United Nations in post-war Iraq." The extractions should be: "thrusting into, agreed, role, war". To address both the general and special cases, we designed two separate prompts containing 7 and 6 demonstrations, respectively. These prompts help LLMs better comprehend a wider range of semantic contexts. The prompt template is provided in Appendix 4.5. In addition, the "pronoun cases" refer to sentences where the trigger is expressed through a pronoun, such as "it" in the example: "Yeah, I

heard something about it." A prompt with 8 demonstrations is delicately designed(The instruction part is in Appendix 4.5).

In this step, we encounter two main challenges. The first is the computationally expensive and time-consuming nature of processing large volumes of data three times within the training dataset. To mitigate this issue, we propose skipping the current LLM extraction step for instances where all label triggers have already been successfully extracted by earlier prompts. The second challenge relates to inconsistencies in tense or singular/plural forms between the extracted triggers and the original sentences(especially Qwen3-1.7B). To address this, we implement an etymological repair mechanism based on a heuristic strategy: if an extracted word does not appear verbatim in the sentence but shows at least $40\%$ character similarity to a word in the sentence, we compare the etymons of both words. If the etymons are identical, the candidate trigger is replaced with the corresponding word from the original sentence. This approach reduces the time complexity from $O(N^2)$ to a more acceptable level.

### 3.1.2 SUPPLEMENTARY EXTRACTION MODULE

After previous step, we identify two common scenarios that would lead to incomplete candidate extraction. First, LLMs may over-attend to certain parts of speech while overlooking other identically critical elements in semantics. For instance, nouns might be extracted while significant verbs are neglected, or vice versa. Consider the sentence: "North Korea on Sunday rejected the U.N. Security Council's plan to discuss the standoff over its suspected nuclear weapons development, calling it 'a prelude to war.'" The initial extraction yielded "nuclear, war, rejected, standoff", but missed a key trigger: "discuss". Second, when a sentence contains multiple event statements linked by subordinate clauses or adverbial phrases, LLMs may fail to capture crucial candidate triggers. For example, in the sentence: "It was a false choice to debate whether Iraq should be run by coalition forces or the United Nations, said Blair, who was believed to be in favor of a stronger UN role in post-conflict Iraq than Bush." The previous extraction included "said, UN, coalition, choice, run, was, debate, United Nations", but omitted the trigger "conflict" and candidate "believed". To mitigate these issues and improve the coverage of golden triggers, we designed two additional prompts specifically targeting the above two scenarios(Appendix 4.5).

### 3.1.3 ITERATIVE REFINEMENT MODULE

After the previous two sub-modules, we have observed that certain important triggers, particularly those expressed as adverbial phrases or embedded within intricate clauses, are still missed(In the sentence: "The scientific conference, attended by Nobel laureates and plagued by heated debates over ethics, reached a consensus with the lead researcher **releasing** a dataset that overturned prior assumptions." The significant adverbial phrase "releasing" was overlooked). To address this limitation, we design three new prompts containing samples with complex syntactic and semantic structures, which are sequentially inferred by LLMs with each input built upon the previous outputs. The three prompts, containing 6, 4, and 5 demonstrations respectively, possess the same prompt template(presented in Appendix 4.5).

The above whole sub-modules fully leverage the hierarchical, step-wise methodology inspired by 'simple-to-complex' scaffolding approach, which expands the coverage of potential candidate triggers as a result. Within the cascading scheme, every extraction step can cover missing authentic triggers that previous steps lost (Experiment 4.4 validates this point). In addition, this method avoids chaotic prompt design, offering a more structured and logical alternative to arbitrary prompt construction.

### 3.2 CHROMA DATABASE CONSTRUCTION

This section proposes the construction of two chroma databases using the training dataset, with the goal of fully leveraging both sentence-level semantic knowledge and word-level contextual knowledge to filter

out irrelevant candidate triggers when processing new data. During the retrieval process, seven predefined samples are drawn from the word contextual chroma database and three samples are from the sentence chroma database. Hyperparameters are then used to combine the cosine similarities from both sources, just as Equation 1 illustrates. Specifically, for samples retrieved from the word contextual database, the cosine similarity at the sentence level needs to be computed sequentially to form the composite score. Similarly, for samples where word-level information is not stored from the sentence chroma database, the cosine similarity of relevant word pairs is also calculated and incorporated.

$$\text{Score} = \alpha \cdot \text{Cosine\_similarity}_{\text{words}} + (1 - \alpha) \cdot \text{Cosine\_similarity}_{\text{sentence}} \tag{1}$$

In addition, previous sections have illustrated that the extractions will inevitably contain arguments and other irrelevant words. Due to the inherent semantic relations among all words in one sentence Vaswani et al. (2017), some irrelevant words or arguments may exhibit strong semantic connections to triggers. This hampers the effective filtration of some negative candidates that possess high semantic similarities with real retrieved triggers. To mitigate this issue, we propose treating arguments and negative candidate triggers in training set as false samples and labeling them with event type "trigger:None", which thereby favors the removal of these types. Furthermore, to enhance the attention on extracted words or triggers in relevant sentences, our method repeats each of these words six times (performance is the best by repeating six times) and places them in the beginning of original sentences, separated by "<SEP>". We use mean embeddings of these words within sentences (skipping the six-times prefixes) combined with metadata to construct Word Contextual Chroma Database. Only sentences containing real triggers are stored in Sentence Chroma Database in the form of mean embeddings, with their triggers as one component of metadata.

### 3.3 TRIGGERS JUDGMENTS AND CLASSIFICATIONS

This part contains threshold filtering, two rule-based filtering, along with an automated filtering process powered by Qwen3-1.7B. The threshold filtering strategy excludes retrieval samples with low similarity scores(0.7). Rule-based filtering methods are designed to further eliminate unreasonable retrieved samples. The first rule is to filter candidate triggers for which the number of distinct clusters exceeds 4. It is motivated by the thought that if a candidate trigger exhibits semantic relationships with multiple event types, this candidate should semantically not belong to any specific event type. The second rule is to remove candidate triggers for which the event type of the top-1 retrieval is "trigger:None". This helps eliminate cases that are highly related to arguments or negative triggers presented in the training set. The retrieved samples will be fixed after two filtering processes. Then, if all the retrievals are belonging to one event type, it indicates that the filtering rules applied earlier are sufficiently confident to regard the candidate as a valid trigger. In this case, the candidate is predicted to be a trigger and assigned the same event type as the retrieved instances, without further LLM judgment. Else, the prompts will be generated automatically and sent to Qwen3-1.7B iteratively to acquire the final inferences. The iterative process is based on sorted clusters(sort by element numbers within clusters) formed by different event types from filtered retrievals.

### 3.3.1 AUTOMATED PROCESS OF PROMPT GENERATION AND LLM JUDGMENT

The main underlying idea of this innovative process is to generate positive and negative samples within prompts automatically on the basis of event-type clusters, and iteratively inferred by LLM to obtain the final judgment, as outlined in Algorithm 1: First, the prompt includes one instruction and two fixed examples in advance to help Qwen3-1.7B grasp basic task motivation(line 1). In an iterative process, from the largest cluster to the smallest, we automatically revise and augment the prompt with additional examples (lines 2-18). For each current cluster, the event type, along with its corresponding sentences and triggers, are treated as positive samples (lines 5-7). Negative samples are drawn from all clusters ranked lower than the current one (lines 8-12). Thus, the final prompt comprises one instruction, two predefined examples, and

automatically selected positive and negative examples, which are sequentially sent to LLM. If the output of LLM is "Yes", this candidate will be predicted as one trigger and assigned the event type of current cluster (lines 14-16). Otherwise, the process continues to the next cluster with the prompt dynamically updated. If, after iterating through all clusters, the candidate is not associated with any event type, it is discarded(line 19). Because through the inference of LLM, the most similar sentences and their event types are not considered relevant to the candidate, indicating that this candidate does not possess enough semantic information to become a trigger and is therefore considered invalid.(The automated prompt is in Appendix 4.5)

---

**Algorithm 1** Automated Prompt Design for Trigger Judgment

---

**Require:** Clustered Elements $C = \{c_1, c_2, ..., c_m\}$, Corresponding Elements from Sorted Cluster $c_i = \{e_1, e_2, ..., e_k\}$, Main Metadata from Element $e_i = \{sentence_i, trigger_i, event\_type_i\}$, Predefined Instruction $p_{inst}$, Predefined Two Demonstrations $p_{demon}$

**Ensure:** True or False

1: p = $p_{inst} + p_{demon}$
2: **for** $l = 1 \rightarrow m$ **do**
3:     $c_i = c_l$
4:     $p_{temp} = ''$
5:     **for** $e_k \in c_i$ **do**
6:         $p_{\text{temp}} + = \text{form\_positive\_prompt\_demons}(sentence_k, trigger_k, event\_type_k)$
7:     **end for**
8:     **for** $c_j = l + 1 \rightarrow m$ **do**
9:         **for** $e_k \in c_j$ **do**
10:             $p_{\text{temp}} + = \text{form\_negative\_prompt\_demons}(sentence_k, trigger_k, event\_type_k)$
11:         **end for**
12:     **end for**
13:     $p + = p_{temp}$
14:     **if** LLM_Judgment(p) **then**
15:         **return** True
16:     **end if**
17:     $p - = p_{temp}$
18: **end for**
19: **return** False

---

## 4 EXPERIMENT

### 4.1 EXPERIMENTAL SETUP

Our experiments are conducted on ACE-05 dataset, which was first introduced by Ahn (2006). We follow the same data split as Lin et al. (2020). The dataset comprises 33 distinct event types. For evaluation, we use the strict matching F1 score for both trigger identification and trigger classification. The hyperparameter $\alpha$, which controls the weight of cosine similarity between dual semantics, is set to 0.8. The score threshold is set to 0.7. Additionally, the number of prefix words that were added before the head position of sentences is set to 6(Comparison of different prefix numbers is in Appendix 4.5).

### 4.2 COMPARED BASELINES

We compare our experimental results with several state-of-the-art generative approaches, including three mainstream generative model-based methods, Text2Event, DEGREE, and DICE, as well as the LLM-based pipeline system STAR.

- **Text2Event** Lu et al. (2021) A controllable event extraction framework implemented via a generative approach, with control achieved through a trie-based constrained decoding algorithm and curriculum learning.
- **DEGREE** Hsu et al. (2021) A template-enhanced generative modeling approach enables the extraction of event triggers and arguments with fewer samples.
- **DICE** Ma et al. (2023) A generative model with template which is similar to DEGREE but employs distinct queries for different argument roles.
- **STAR** Ma et al. (2024) A structure-to-text data generation framework accompanied by a self-refinement strategy for in-context-learning by LLMs.

## 4.3 MAIN RESULTS

Table 1 presents a comparative analysis between our method and other baselines. The best and second-best results were both achieved under our proposed framework. Our top-performing model, based on Qwen3-4B, surpasses the previous state-of-the-art by $2.78\%$ in trigger identification and $4.74\%$ in trigger classification. With the exception of identification performance based on Llama2-7B model, which falls slightly below the current best, all the other seven metrics evaluated exceed the state-of-the-art levels. These results demonstrate the effectiveness of our novel approach. The significant improvement in classification performance, in particular, highlights the benefit of incorporating the dual retrieval mechanism. Furthermore, the superior performance of the much smaller Qwen3-1.7B compared to GPT-3.5 demonstrates the high effectiveness of our method. We also observed the performance gap between Qwen3 and Llama2, suggesting the superior comprehension capabilities of Qwen3.

Table 1: Main Results. Baselines (lines 2-7) use GPT-3.5 and GPT-4.0 to generate data within the STAR architecture.

| Dataset | Models | Method | Performance (%) | |
|---|---|---|---|---|
| | | | Trigger Identification | Trigger Classification |
| ACE-05 | GPT-3.5 | Text2Event | 11.30 | 3.47 |
| | | DEGREE | 17.52 | 6.21 |
| | | DICE | 16.94 | 7.09 |
| | | Instruction | 18.31 | 8.37 |
| | | Instruction+Examples | 59.71 | 53.29 |
| | GPT-4.0 | Instruction+Examples | 62.12 | 56.46 |
| ACE-05 | Llama2-7B | REVO-ED | 61.92 | 58.25 |
| | GPT-3.5 | | 63.01 | 58.13 |
| | Qwen3-1.7B | | 63.75 | 59.93 |
| | Qwen3-4B | | **64.90** | **61.20** |

## 4.4 IMPORTANCE OF EXTRACTION PROMPT

To assess the performance of our proposed candidate extraction pipeline, we store all extraction results across sub-modules and count the number of samples in which the golden triggers are not identified, shown as a line graph in Figure 2. We can observe that Llama2-7B performs well in the Basic Extraction Module but its extraction coverages in three refinement steps are nearly not changed, which seems to imply that Llama2-7B's ability in semantic comprehension is great, but it cannot comprehend the demonstrations in complicated semantic scenarios, which restricts its final performance. In contrast, the comparatively opposite

change performed in Qwen3-1.7B appears to indicate its distinguished comprehension in examples but low capacity in understanding instructions. The performance of GPT-3.5 is satisfactory, but its final result is slightly inferior to Qwen3-1.7B. The outstanding performance in Qwen3-4B validates its superior ability under our scheme. Furthermore, from the above comparison and the experimental results, we can conclude that there is a positive correlation between the final evaluation metrics and the quality of the extracted results, demonstrating the effectiveness of our fine-grained retrieving process.

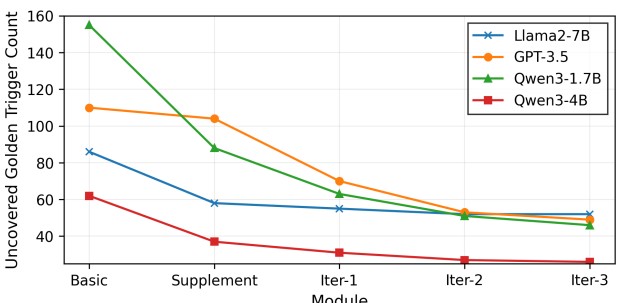

Figure 2: Uncovered Golden Trigger Count by Every Extraction Prompt Step. "Basic" column means the Basic Extraction Module and the "Sup- plement" indicates the Supplementary Extraction Module. The "Iter-1", "Iter-2", and "Iter-3" individually represents per loop in the Iterative Refinement Module.

## 4.5 ABLATION STUDY

We conducted six ablation experiments to validate the effectiveness of each component in the scheme, as illustrated in Table 2. Significant performance declines were observed when removing word embeddings, the score threshold, or negative sample filtering, confirming their indispensable roles in the framework. These components represent crucial and innovative contributions of our work. A minor decrease in performance after removing sentence embeddings suggests that relying solely on word-level textual semantics is insufficient for optimal retrieval, which confirms the effectiveness of semantics in sentences to some extent. The slight drop upon removing Cluster Count Limitation indicates its certain importance in the filtering process. We assume its special influence in scenarios that involve large-scale retrieval sets with high semantic diversity and multiple event types. The improvement achieved by LLM Automated Filtering highlights the value of LLMs in the final determination and the effectiveness of this automated process.

Table 2: Ablation Study Results on ACE-05 Dataset using Qwen3-1.7B Model

| Configuration | Trigger ID | Trigger CLS |
|---|---|---|
| REVO-ED (Full Model) | 63.75 | 59.93 |
| *Ablation Studies:* | | |
| w/o Sentence Embeddings | 60.69 | 56.68 |
| w/o Word Embeddings | 18.54 | 12.23 |
| w/o Score Threshold Filtering | 31.26 | 30.60 |
| w/o Cluster Count Limitation | 63.74 | 59.92 |
| w/o Negative Cases Filtering | 27.50 | 25.64 |
| w/o LLM Automated Filtering | 62.71 | 58.98 |

ACKNOWLEDGMENTS

This work was supported by the Natural Science Foundation of China (No. 62372057), and the Key Laboratory of Trustworthy Distributed Computing and Service (MOE).

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

APPENDIX

## A    The Prompts of Candidate Extraction

### A.1    Basic Extraction Module

**A.1.1    Simple Cases and Special Cases**    The prompt of extracting simple cases and special cases in the Basic Extraction Module is as follows:

**Prompt of Simple Cases and Special Cases**

```
<|im_start|>[user]:
{prompt}
Extract the event triggers in the input sentence. An event trigger is an
    indicative word or phrase in a sentence that clearly signals the
    occurrence of one event. It can be a verb (e.g., "explode"), A noun (e.g
    ., "summit"), or A pronoun (e.g., "it", "this"), and it can also be a
    phrase (e.g., "War World II") or A combination of words with different
    parts of speech (e.g., "shoot at", "took over"). For each input sentence,
     determine whether it describes one or more events or contains event
    information(The more events in the sentence can be mentioned in clauses
    or conjunctions, etc). If there exists one or more events, extract the
    corresponding event triggers. If no event trigger exists or the sentence
    doesn't contain event information, output :"trigger 1: None". Otherwise,
    output the one or more event triggers in the input sentence in the form
    of the following output format.
Output Format:
If no event trigger exists, output:
trigger 1: None
Otherwise, list all triggers in order:
trigger 1: XXXX
(trigger 2: XXXX)
(trigger 3: XXXX)
(...)
Input Examples:
...
<|im_end|>
<|im_start|>assistant
<th>think>
```

**A.1.2 Pronoun Cases** The prompt of extracting pronoun cases in the Basic Extraction Module is as follows:

**Prompt of Pronoun Cases**

```
<|im_start|>[user]:
{prompt}
Find implicit event-triggering pronouns (it, this) in the given sentence. The
    pronouns should semantically refer to an event. If the required pronouns
    don't exist, output :"trigger 1: None". Otherwise, output the pronoun in
    the form of the following format.
Output Format:
If no implicit event-triggering pronouns exist, output:
trigger 1: None
Otherwise, output the implicit event-triggering pronouns:
trigger 1: XXXX
Input Examples:
...
<|im_end|>
<|im_start|>assistant
<th>think>
```

## A.2 The Prompt of Supplementary Extraction Module

**A.2.1 Missed Events Retrospections** The prompt of sub-module - Missed Events Retrospections - in the Supplementary Extraction Component is as follows:

---
**Prompt of Missed Events Retrospections**

```
<|im_start|>[user]:
{prompt}
Extract the potential unextracted event structural words that are refering to
    the given event words(splitted by ",") in the input sentence. Such
    unextracted event structural words are important in indicating an event
    information. The corresponding number of such structural information may
    be one or more. Output the potential unextracted structural information
    in the following output format.
Output Format:
Information 1: XXXX
(Information 2: XXXX)
(Information 3: XXXX)
(...)
Input Examples:
...
<|im_end|>
<|im_start|>assistant
<th>think>
```
---

**A.2.2 Event Information Completion** The prompt of sub-module - event information completion - in the Supplementary Extraction Component is:

---
**Prompt of Event Information Completion**

```
<|im_start|>[user]:
{prompt}
Extract the event structural words that are not refering to but possess
    connections with the given event words(splitted by ","). Such event
    structural words are important in indicating an event information. The
    corresponding number of such structural information may be one or more.
    Output these event structural information in the following output format.
Output Format:
Information 1: XXXX
(Information 2: XXXX)
(Information 3: XXXX)
(...)
Input Examples:
...
<|im_end|>
<|im_start|>assistant
<th>think>
```
---

**A.3 The Prompt of Iterative Refinement Module** The prompt for the Iterative Refinement Module is as follows:

**Prompt of Iterative Refinement Module**

```
<|im_start|>[user]:
{prompt}
It is known that event structural words are important informations that can
    indicate an event occurrence. Given the known structural words of one
    sentence, there may still exsit some potentially unextractced event
    structural words. The corresponding number of such unextracted structural
     words may be one or more. Extract them and output them in the following
    output format.
Output Format:
Information 1: XXXX
(Information 2: XXXX)
(Information 3: XXXX)
(...)
Input Examples:
...
<|im_end|>
<|im_start|>assistant
<th>think>
```

**B   The Automated Prompt Design**   The automated prompt design by fined-graind LLM judgment
is as follows:

**Prompt of Automated Prompt Design**

```
<|im_start|>[user]:
{prompt}
Given an event statement and its corresponding event trigger word, where an
    event trigger refers to an indicative word or phrase that signals the
    occurrence of a specific event.Please determine whether the trigger word
    belongs to the specified event type based on its contextual meaning and
    the overall semantics of the statement. If it belongs, respond with "yes
    "; otherwise, respond with "no". Then, explain your answer.
Output Format:
Yes./No.
Input Examples:
### Input 1
Sam Waksal, founder of the US pharmaceutical company ImClone Systems was
    sentenced to 87 months in prison Tuesday for insider trading.
### Trigger
sentenced
### Event Type
Justice:Sentence
### Answer
Yes.
...
<|im_end|>
<|im_start|>assistant
<th>think>
```

**C** **Experimental Results with different prefix number**    We conduct a series of experiments to evaluate the impact of varying the number of prefix candidates during the construction of the Word Contextual Chroma Database based on Qwen3-1.7B model. The F1-score results for different trigger numbers (ranging from 0 to 8) are presented in the Table 3. The first column indicates different prefix counts, while the second and third columns present the corresponding results and comparisons between the optimal outcome (achieved with 6 prefixes) and those obtained with other counts up to 8. Minor variations are observed across prefix counts ranging from 0 to 7, with fluctuations within -1.10 to -0.31 for identification and -0.92 to -0.29 for classification, demonstrating the robustness of our approach to various prefix quantities. However, beyond 7 prefixes, both performances decline sharply, with 36.31 decline in identification and 27.04 deduction in classification compared to the best results. We speculate that an excessive number of prefix words may disrupt the original sentence semantics, resulting in the performance decline of retrieval process.

Table 3: Compared Results of Trigger Identification and Classification with Different Prefix Number When Constructing Word Contextual Chroma Database Based On Qwen3-1.7B

| Diff. Pre. Num. | Trigger ID (%) | Trigger CLS (%) |
|---|---|---|
| 6 (Optimal Results) | 63.75 | 59.93 |
| *Compared Results* | | |
| 0 | 62.80 (-0.95) | 59.01 (-0.92) |
| 1 | 62.65 (-1.10) | 59.17 (-0.76) |
| 2 | 62.88 (-0.87) | 59.64 (-0.29) |
| 3 | 63.40 (-0.35) | 59.40 (-0.53) |
| 4 | 63.22 (-0.53) | 59.45 (-0.48) |
| 5 | 63.44 (-0.31) | 59.64 (-0.29) |
| 7 | 62.91 (-0.84) | 59.31 (-0.62) |
| 8 | 27.44 (-36.31) | 27.04 (-32.89) |