# OpenReview forum: "REVOLUTIONIZING EVENT DETECTION: A NOVEL PROMPT-DRIVEN METHOD ENHANCED BY RETRIEVAL-AUGMENTED PARADIGM"
_ICLR.cc/2026/Conference — Submitted to ICLR 2026_

### Official Review · Reviewer_Jaj5 · 2025-10-27

**Soundness:** 2
**Presentation:** 1
**Contribution:** 2
**Rating:** 4
**Confidence:** 3

**Summary:**

The author proposes a novel retrieval-augmented prompt-driven method that integrates a retrieval augmentation mechanism with a redesigned cascaded prompt-driven framework for sentence-level event extraction. The proposed method demonstrates relatively good performance even on smaller parameter large language models.

**Strengths:**

1 . The proposed method demonstrates relatively good performance even on smaller parameter large language models.

**Weaknesses:**

Weaknesses:

1. The task conducted remains one of the most fundamental tasks in event extraction. To my knowledge, event extraction in the ACE dataset not only includes trigger word extraction but also involves entities and the matching of entities with events. Merely extracting trigger words and identifying their types may not hold significant meaning after the emergence of large models, and the author may have overstated the contribution's value.

2. Although the ACE dataset is a benchmark for event extraction, it is still limited to sentence-level event extraction and is relatively outdated. Relying solely on one dataset to validate the method appears insufficient.

3. It remains unclear whether the proposed method is applicable to paragraph-level event extraction.

4. To my knowledge, without using large models, the accuracy of fine-tuned smaller models on the ACE dataset has already reached around 75. The author's results, however, are too low. For instance, the original DICE paper reported F1 scores of 75.22 and 70.46 for trigger word identification and classification, respectively, on the ACE dataset. In contrast, the author's results are only in the teens, which seems excessively low and warrants further scrutiny.

5. The overall experimental design is overly simplistic, making it difficult to fully demonstrate the effectiveness of the proposed method. How does the performance of the comparative methods compare to fine-tuned non-model-based methods?

**Questions:**

see Weaknesses

---

### Official Review · Reviewer_L64g · 2025-10-30

**Soundness:** 2
**Presentation:** 1
**Contribution:** 2
**Rating:** 2
**Confidence:** 4

**Summary:**

This paper presents REVO-ED, an event detection framework combining a multi-step cascading prompt strategy with a dual retrieval mechanism, aiming to reduce manual prompt design and improve scalability.

**Strengths:**

1. Introduces an automated ED approach that alleviates per-event-type manual prompt design, improving scalability.

2. Achieves competitive performance with lightweight LLMs on ACE-05.

**Weaknesses:**

1. All experiments are conducted solely on ACE-05 (2006), which is relatively small and outdated, lacking validation on larger, more complex, or cross-domain datasets.

2. The proposed framework essentially combines multiple LLM components into a multi-step pipeline, with several empirically chosen hyperparameters (e.g., the number of prefix words set to 6, similarity threshold set to 0.7) derived purely from observation. This raises concerns about the need to re-tune parameters for different datasets, given the absence of theoretical justification or adaptive mechanisms.

3. The current setup involves only 33 event types. Since the method claims efficiency compared to per-type prompt design, experiments with significantly more event types would help validate this advantage.

4. The retrieval database is constructed from the training set. If test samples are semantically close to training sentences, performance might be inflated, potentially limiting generalization.

**Questions:**

N/A

---

### Official Review · Reviewer_8yGw · 2025-10-31

**Soundness:** 2
**Presentation:** 2
**Contribution:** 2
**Rating:** 2
**Confidence:** 4

**Summary:**

This paper proposes a new prompt-based method for event detection. Specifically, this paper addresses a key limitation of previous work: as the number of event types grows, the number
of prompts needed increases linearly, resulting in significant manual effort and computational costs. Therefore, this paper propose a new method which integrates a retrieval-augmented mechanism with a redesigned cascading prompt-based framework. This method first extracts candidate trigger and then use the retrieval-augmented module to filter irrelevant candidates. Finally, this method adopts a prompt-based method to accurately match valid triggers with their corresponding event types based on retrieved information. Experimental results on ACE 2005 demonstrates the effectiveness of the proposed method.

**Strengths:**

1. The topic focused in this paper is meaningful. Enabling LLMs to automatically extract events is highly meaningful for advancing information extraction and deep information seeking (such as DeepResearch).
2. The presentation of the paper is clear and easy to follow.

**Weaknesses:**

1. The proposed method mainly relies on prompt engineering for a specific task, without demonstrating its generalizability. For such tasks, a fine-tuned BERT model often performs quite well, which raises concerns about the paper’s fundamental contribution and technical novelty.
2. The experiments are not sufficiently comprehensive, as evaluation is conducted only on the ACE 2005 dataset. It would be more convincing to include additional benchmarks such as TACKBP or RAMS to better validate the method’s effectiveness. Furthermore, prior studies have found limitations in the evaluation of existing IE datasets [1]. It is unclear whether the authors have considered these issues. The paper would also benefit from comparisons with smaller, task-specific fine-tuned models, such as BERT, RoBerta, etc.

[1] Xu, Derong, et al. "Large language models for generative information extraction: A survey." *Frontiers of Computer Science* 18.6 (2024): 186357.

**Questions:**

See Weaknesses.

---

### Official Review · Reviewer_hzQF · 2025-11-02

**Soundness:** 2
**Presentation:** 2
**Contribution:** 2
**Rating:** 2
**Confidence:** 5

**Summary:**

The paper proposes a cascade prompting and retrieval-augmented pipeline for event detection with LLMs. It over-generates candidate triggers using generic prompts, then filters and assigns event types via dual-granularity vector retrieval with lightweight LLM judgments. On ACE-05, the method reports state-of-the-art results without task-specific fine-tuning and with a small prompt budget. The work is positioned for resource-constrained, practical deployments.

**Strengths:**

1. This paper presents a successful application of LLMs to the ED task.
2. The proposed methods outperform previous LLM-based approaches and are practical for real-world use.

**Weaknesses:**

1. The model was only tested on a single dataset. It's unclear if it actually works in other situations or if it can handle unexpected inputs.

2. There was basically no new training or fine-tuning, and the improvements are tiny. This "innovation" sounds more like just optimizing the workflow rather than creating a genuinely new model.

4. The range of problems it solves is too narrow, and it doesn't even include argument extraction, which is a pretty important part of event extraction field.

5. The paper doesn't compare it to the latest top-tier models (like GPT-5 or Gemini 2.5 Pro), which makes its experimental results less convincing.

6. It also skips comparisons with older, smaller supervised models, so it's hard to tell if this new approach is even better than those.

**Questions:**

See weakness

---

### Meta-Review · Area_Chair_ZM26 · 2026-01-03

**Summary:**

Reviewers highlighted concerns including limited evaluation on a single outdated dataset (ACE-05), lack of generalizability and comparisons to fine-tuned small models or latest LLMs, empirical hyperparameters without theoretical backing, narrow task scope (e.g., no argument extraction or paragraph-level applicability), potential performance inflation from training-set retrieval, and an anonymity violation in the acknowledgments, informing a suggested rejection due to insufficient novelty, rigor, and ethical compliance.

**Reviewer Concerns:**

No rebuttal is present in the provided document, so none of the concerns—such as dataset limitations, missing comparisons, methodological justifications, scope narrowness, and the anonymity issue—have been addressed; all remain outstanding.

**Reviewer Scores:**

Without a rebuttal or further discussion, reviewers would likely retain their original scores.

---

### Decision · Program_Chairs · 2026-01-26

Reject